# Problems of magnetic resonance diagnosis for gastric-type mucin-positive cervical lesions of the uterus and its solutions using artificial intelligence

**Ayumi Ohya[1], Tsutomu Miyamoto [2]\*, Fumihito Ichinohe[1], Hisanori Kobara[2], Yasunari Fujinaga[1], Tanri Shiozawa[2]**

1 Department of Radiology, Shinshu University School of Medicine, Matsumoto, Japan, 2 Department of Obstetrics and Gynecology, Shinshu University School of Medicine, Matsumoto, Japan

\* tmiya@shinshu-u.ac.jp

**Data Availability Statement:** The data that support the findings of this study are available in Shinshu

## Abstract

### Purpose

To reveal problems of magnetic resonance imaging (MRI) for diagnosing gastric-type mucin-positive (GMPLs) and gastric-type mucin-negative (GMNLs) cervical lesions.

### Methods

We selected 172 patients suspected to have lobular endocervical glandular hyperplasia; their pelvic MR images were categorised into the training (n = 132) and validation (n = 40) groups. The images of the validation group were read twice by three pairs of six readers to reveal the accuracy, area under the curve (AUC), and intraclass correlation coefficient (ICC). The readers evaluated three images (sagittal T2-weighted image [T2WI], axial T2WI, and axial T1-weighted image [T1WI]) in every patient. The pre-trained convolutional neural network (pCNN) was used to differentiate between GMPLs and GMNLs and perform four-fold cross-validation using cases in the training group. The accuracy and AUC were obtained using the MR images in the validation group. For each case, three images (sagittal T2WI and axial T2WI/T1WI) were entered into the CNN. Calculations were performed twice independently. ICC (2,1) between first- and second-time CNN was evaluated, and these results were compared with those of readers.

### Results

The highest accuracy of readers was 77.50%. The highest ICC (1,1) between a pair of readers was 0.750. All ICC (2,1) values were <0.7, indicating poor agreement; the highest accuracy of CNN was 82.50%. The AUC did not differ significantly between the CNN and readers. The ICC (2,1) of CNN was 0.965.

University Institutional Repository at [http://hdl.handle.net/10091/0002000343].

**Funding:** T.M., A.O., H.K., and T.S. have received funding from the Japan Society for the Promotion of Science (JSPS) KAKENHI, Grant Number 22K09593 (https://kaken.nii.ac.jp/ja/grant/KAKENHI-PROJECT-22K09593/). JSPS did not play any role in the study design, data collection and analysis, decision to publish, or preparation of the manuscript?

**Competing interests:** The authors have declared that no competing interests exist.

## Conclusions

Variation in the inter-reader or intra-reader accuracy in MRI diagnosis limits differentiation between GMPL and GMNL. CNN is nearly as accurate as readers but improves the reproducibility of diagnosis.

## Introduction

Common benign lesions in the uterine cervix include Nabothian cysts, tunnel clusters, lobular endocervical glandular hyperplasia (LEGH), endometriosis, and cervical polyps [1]. LEGH is a benign lesion first proposed by Nucci et al. [2], but it may be a precursor lesion for gastric-type mucinous carcinomas (GAS) [3–8]. Therefore, frequent and long-term follow-up or surgical treatment is selected once LEGH is diagnosed. In contrast, other benign cystic lesions of the uterine cervix do not require follow-up as frequently as LEGH. Therefore, clinically, LEGH and other benign cystic lesions must be distinguished.

LEGH shows magnetic resonance imaging (MRI) findings called 'cosmos pattern', while Nabothian cysts show coarse cysts pattern [9]. However, some Nabothian cysts exhibit MRI findings similar to those of LEGH [10]. The decisive difference from other benign lesions is that LEGH and GAS secrete gastric-type mucin, which has *O*-linked oligosaccharides with a terminal α1,4-linked *N*-acetylglucosamine (αGlcNAc) residue [11]. Because gastric-type mucin is a neutral mucin, LEGH and GAS exhibit a 'two-color pattern' on Pap smears [12]. Further, αGlcNAc has been detected in cervical mucus by latex agglutination assay using monoclonal antibody HIK1083 [13]. This method has extremely high sensitivity and specificity [13]; however, the number of facilities that can implement this method is limited. Therefore, differentiation by MRI findings is extremely important.

An attempt was recently made to classify cervical lesions into gastric-type mucin-positive lesions (GMPLs) and gastric-type mucin-negative lesions (GMNLs) based on MRI findings [10]. The specificity was 95.5% when the cosmos pattern was observed as a hypointense area compared with the cervical stroma on T1-weighted images (T1WIs) [10]. However, the accuracy of differentiating GMPLs from GMNLs by MRI findings is not clear. In addition, no study of differences in GMPLs diagnostic performance among physicians has been reported.

On the other hand, the field of machine learning has developed remarkably. A convolutional neural network (CNN) is a machine learning algorithm of great interest in the field of diagnostic imaging [14]. It can perform equivalently to or better than humans in some image classification tasks [14]. In particular, transfer learning using pre-trained CNNs (pCNNs) can achieve high classification performance with a relatively small dataset [14]. Although GMPLs, as represented by LEGH, are relatively infrequent, it is not impossible to diagnose them using the pCNN. In addition, since artificial intelligence always makes the same diagnosis once it learns, it can be a promising solution when the reproducibility of physicians' diagnoses is low. However, transfer learning has randomness in its learning, and the degree of reproducibility of the learning results has not been fully investigated.

Therefore, this study aimed to clarify the accuracy of MRI diagnosis of GMPLs, the differences in diagnostic ability among physicians, and the problems in the current situation, and to explore the possibility of GMPLs diagnosis by artificial intelligence.

## Material and methods

### Patient population

We reviewed the medical records in our hospital and selected 172 consecutive patients with clinical suspicion of LEGH or GAS (based on ultrasonographic findings, such as multiple cysts of the cervix, and symptoms, such as vaginal watery discharge) who underwent pelvic MRI between January 2000 and October 2020. Patients ranged in age from 26 to 82 years, with an average age of 48.7 years. The process of patient selection, and grouping is summarized in **Fig 1**. In 171 patients, a cervical Pap smear or latex agglutination assay using monoclonal antibody HIK1083 (HIK test) (Cica HIK gastric-type mucin; Kanto Kagaku, Tokyo, Japan) [13] had been performed to confirm that the cervical mucus included gastric-type mucin. Among the 172 patients, 35 underwent surgery or biopsy and were pathologically diagnosed with benign cystic lesions other than LEGH (BCL), LEGH, LEGH with atypia or adenocarcinoma in situ (aLEGH), and GAS. One patient who had not undergone a Pap smear and latex agglutination assay before surgery was pathologically diagnosed with GAS after surgical resection. Patients with gastric-type mucin confirmed by a cervical Pap smear or latex agglutination assay and those with LEGH or GAS pathologically diagnosed after surgical resection were both included in the GMPL group. Patients without gastric-type mucin or pathologically diagnosed BCLs were included in the GMNL group. There were 76 and 96 patients in the GMPL and GMNL groups, respectively. Thirty-one of the 76 patients in the GMPL group underwent surgical resection or biopsy. Of these, 15 patients exhibited LEGH, 13 exhibited aLEGH, and three exhibited GAS. In contrast, in the GMNL group, surgical resection was performed in only four of 96 patients. In these four patients, surgical resection was performed for a disease other than cervical lesions. Two of these patients were pathologically diagnosed with Nabothian cysts, and the other two were diagnosed with tunnel clusters. The classification of histopathologic lesions is summarized in Table 1.

This study was approved by the ethics committee of our institution (approval no.: 4423). The ethics committee waived the requirement for informed consent for the use of the patients' information and MR images because diagnostic use of the samples had been completed before the study, and there was no risk to the involved patients. The patients' information and MR images were also coded to protect patient anonymity.

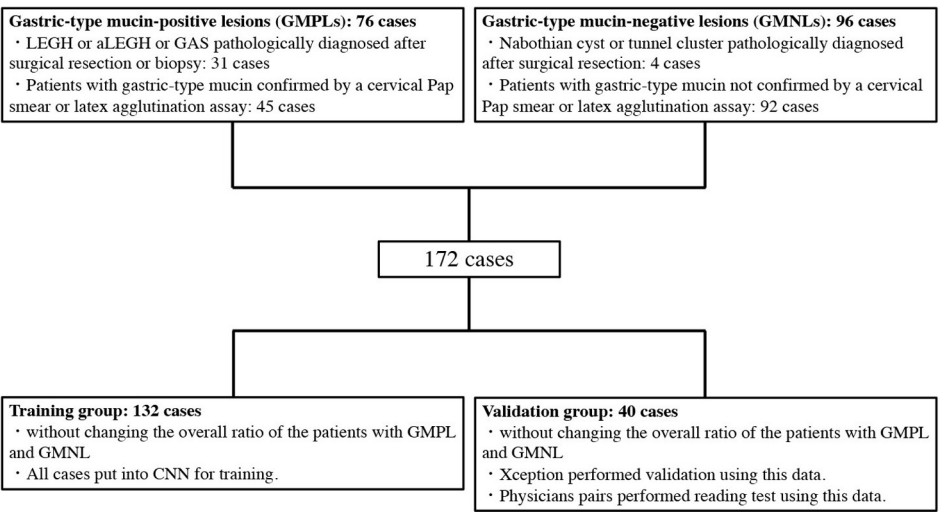

**Fig 1. Patient selection and inclusion criteria and grouping.**

**Table 1. Definition of lesions.**

| Lesion | definition | n | Mean age |
|---|---|---|---|
| GMPLs | | | |
| LEGH | Histopathologically diagnosed LEGH, LEGH with atypia, and LEGH with AIS | 28 | 44 years |
| GAS | Histopathologically diagnosed GAS | 3 | 42 years |
| other | Positive gastric type mucin by cervical Pap smear or latex agglutination assay | 45 | 51 years |
| GMNLs | | | |
| Nabothian cyst | Histopathologically diagnosed Nabothian cyst | 2 | 49.5 years |
| Tunnel cluster | Histopathologically diagnosed tunnel cluster | 2 | 51 years |
| other | Nagative gastric type mucin by cervical Pap smear or latex agglutination assay | 92 | 48 years |

GMPLs, gastric-type mucin-positive lesions; GMNLs, gastric-type mucin-negative lesions; LEGH, lobular endocervical glandular hyperplasia; AIS, adenocarcinoma in situ; GAS, gastric-type mucinous adenocarcinoma

Lesions in patients who were determined to have gastric-type mucin secretion by surgery, biopsy, cervical Pap smear, or latex agglutination assay were designated gastric-type mucin-positive lesions (GMPLs). There were 76 GMPLs. Of these, 31 cases were determined by surgery or biopsy (Table 1). The remaining lesions were determined by cervical Pap smear or latex agglutination assay (Table 1). In contrast, lesions in patients who were determined to not have gastric-type mucin secretion by surgery, biopsy, cervical Pap smear, or latex agglutination assay were designated gastric-type mucin-negative lesions (GMNLs). There were 96 GMNLs. Of these, 4 cases were determined by surgery or biopsy (Table 1). The remaining lesions were determined by cervical Pap smear or latex agglutination assay (Table 1). The total number of GMPLs was 172. From the 172 cases, 132 cases were randomly selected for training the convolutional neural network (CNN) without changing the overall ratio of the patients with GMPL and GMNL. The remaining cases were used as data for CNN validation and physician pair reading experiments. Training data are more numerous than validation data because more data are needed to train the CNN.

## MR images

The most recent pelvic MR images of each patient stored on the image server of our institution were used for analysis. If a hysterectomy or conisation was performed, MR images immediately before the treatment were selected. Of these MR images, the sagittal T2-weighted images (T2WIs) with or without fat suppression, axial T2WIs with or without fat suppression, and axial T1WIs with or without fat suppression were used for analysis. The types and magnetic field strength of the MRI unit used for imaging and the parameters of each sequence varied because the study was retrospective and MR images were acquired over a long period. All the patients underwent MRI using 1.5 or 3.0 tesla scanners. A total of 134 patients underwent MRI by Siemens scanners (Siemens Healthcare Diagnostics, Erlangen, Germany)– 28 patients by 3T Trio, 27 patients by 3T Prisma, 22 patients by 3T Vida, 2 patients by 3T Skyra, 39 patients by 1.5T Avanto, 11 patients by 1.5T Symphony, three patients by 1.5T Aera, and two patients by 1.5T Essenza. A total of 29 patients underwent MRI by GE scanners (GE HealthCare, Chicago, Illinois, USA)–two patients by 3T DISCOVERY MR 750w, one patient by 3T SIGNA Pioneer, one patient by 1.5T OPTIMA MR360w, 15 patients by 1.5T OPTIMA MR450w, 10 patients by 1.5T SIGNA HDxt, and one patient by 1.5T SIGNA Excite HD. The remaining nine patients were imaged with two types of 3T MRI scanners (Philips Electronics N.V., Amsterdam, Holland; Canon Medical Systems Corp., Tochigi, Japan) and three types of 1.5T MRI scanners (Philips Electronics N.V., Amsterdam, Holland; Canon Medical Systems Corp.,

Tochigi, Japan). All images had a slice thickness of 2 to 7.5 mm and were captured as two-dimensional images. One image showing the maximum cross-section of the lesion was selected from each of these three sequences per patient, and the three images were used for analysis.

## Diagnostic accuracy and reproducibility by the readers

The patients were randomly divided into the training (132 patients) and validation (40 patients) groups without changing the overall ratio of the patients with GMPL and GMNL (Fig 1). Three pairs of readers [six readers; two experienced gynecologic radiologists (with 31 and 13 years of experience), pair A; two young radiologists (with 8 years of experience), pair B; and two gynecologists (with 14 and 10 years of experience), pair C) diagnosed GMPLs or GMNLs on the MR images in the validation group without the patients' clinical information. At this time, each pair of readers determined the confidence level of the diagnosis as a percentage by consensus. The confidence level of the diagnosis was stated as a percentage based on each physician's experience. Each reader was presented with a total of three images (sagittal T2WI, axial T2WI, and axial T1WI, which were the maximum cross sections of the lesion) in every patient. The diagnostic accuracy and area under the curve (AUC) in each of the three pairs were determined. This evaluation by the readers was performed twice, at least 1 month apart. Images of the same patients were presented to each pair of readers during the second evaluation, but MR images of the patients were presented in a different order from the first evaluation. Same as the first evaluation, the diagnostic accuracy and AUC in each group were determined. In addition, intraclass correlation coefficients (ICC) (1) or (2) value was calculated for the assessment of reproducibility. ICC value of <0.7 was considered as poor agreement.

## Diagnostic accuracy evaluation by pCNNs

Among the existing pCNNs that can be used by anyone, fine-tuning was performed using Xception [15]. Xception is pre-trained by the ImageNet database. Xception was used on MATLAB software (MATLAB 2020a; MathWorks, Natick, MA, USA). We trained Xception using the images of the training group (132 patients) and validated it using the images of the validation group (Fig 1). When images were input into Xception, MR images were processed so that only the lesion and cervical stroma were included in the image range (Fig 2). The image size entered into 299 × 299 pixels for Xception. As the MR images were captured under various conditions, it was impossible to make the display conditions in the image viewer constant. Therefore, the display conditions were adjusted so that the images had visually similar contrast. The fine-tuning hyperparameters were as follows: the optimiser Adam, learning rate of 0.001 for fully connected layers, and learning rate of 0.00005 for other layers. Fine-tuning was performed with a mini-batch size of 32 and an epoch number of 16. The learning rate was

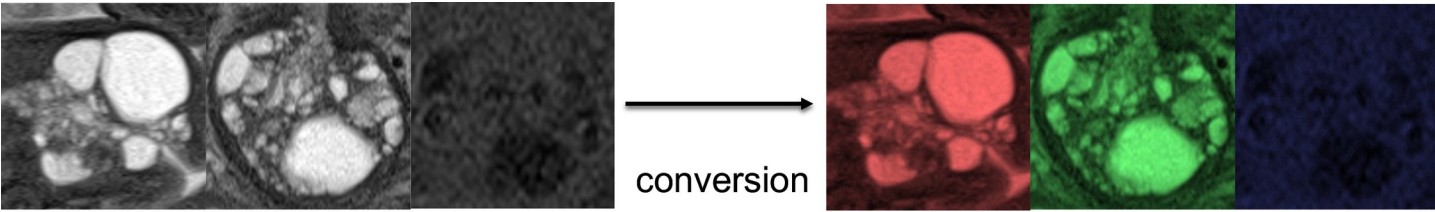

**Fig 2. Three images input into the pre-trained convolutional neural networks.** A sagittal T2-weighted image was input into the red channel, an axial T2-weighted image was input into the green channel, and a T1-weighted image was input into the blue channel. These converted images were processed to include only the lesion and cervical stroma of the uterus.

multiplied by 0.9 every four epochs. For image augmentation, resizing, rotation, translation, and reflection were performed at random. Using the training data, all three images were input into Xception. The training data were randomly divided into four parts without changing the ratio of GMPL to GMNL, and we performed four-fold cross-validation (Fig 3). The accuracy of each verification by the four-fold cross-validation was calculated and averaged to the tentative diagnostic accuracy, which indicates degree of training. Finally, images of the independent validation data were input into the model trained in each fold, and the true diagnostic accuracy and AUC for the independent validation data were determined by averaging the diagnostic probabilities by the four models (Fig 3). These series of calculations were performed twice independently. The ICC (2) value was calculated to evaluate the reproducibility between the two answers.

## Statistical analysis

Creation of a receiver operating characteristic (ROC) curve, calculation of the AUC, comparison of AUCs, and calculation of ICC (1) or (2) values were performed using BellCurve for Excel (Social Survey Information Co., Ltd. Tokyo, Japan). The DeLong's test for two correlated ROC curves was used to compare AUCs. Values of $p < 0.05$ were considered as statistical significance. If the 95% confidence intervals (95% CI) did not overlap, the difference was considered statistically significant.

## Results

### Diagnostic ability of the readers

The diagnostic accuracy, AUC, and ICC (1, 1) for the three pairs of readers are summarised in Table 2. The diagnostic accuracy showed the highest value of 0.775 for the first evaluation of pair B. The diagnostic accuracy showed the lowest value of 0.625 for the first evaluation of pair C. When GMPL was positive in the evaluation with the highest diagnostic accuracy (the first evaluation of pair B), the precision, recall, specificity, and F-measure were 0.737, 0.778, 0.773, and 0.757, respectively.

The AUC of the readers showed the highest value of 0.814 for the second evaluation of pair A. The AUC of the readers showed the lowest value of 0.720 for the first evaluation of pair A.

The highest ICC (1,1) value between the readers in pair C was 0.750 (Table 2). The ICC (2,1) values between the pairs of readers are summarised in Table 3. All ICC (2) values were <0.7, indicating poor agreement.

### Diagnostic ability of Xception and comparison with the ability of the readers

The diagnostic accuracy, AUC, and ICC (2, 1) obtained by Xception are summarised in Table 4. The true diagnostic accuracy of Xception was almost identical to or higher than the tentative diagnostic accuracy in most of the procedures. The diagnostic accuracy showed the highest value of 0.825 for the second time. This was higher than the highest diagnostic accuracy (0.775) of the readers (Table 2). The AUC showed the highest value of 0.854 for the second time. Comparison of the ROC curve between the AUC values of Xception and those of each pair of readers showed no statistically significant difference (Fig 4). When GMPL was positive in the procedure with the highest diagnostic accuracy, the precision, recall, specificity, and F-measure were 0.867, 0.722, 0.909, and 0.788, respectively. All of these values except recall were higher than those of the readers. The ICC (2,1) value (95% CI) was 0.965 (0.934–0.981). This value was higher than all ICC (2,1) values of the readers. The agreement rate was

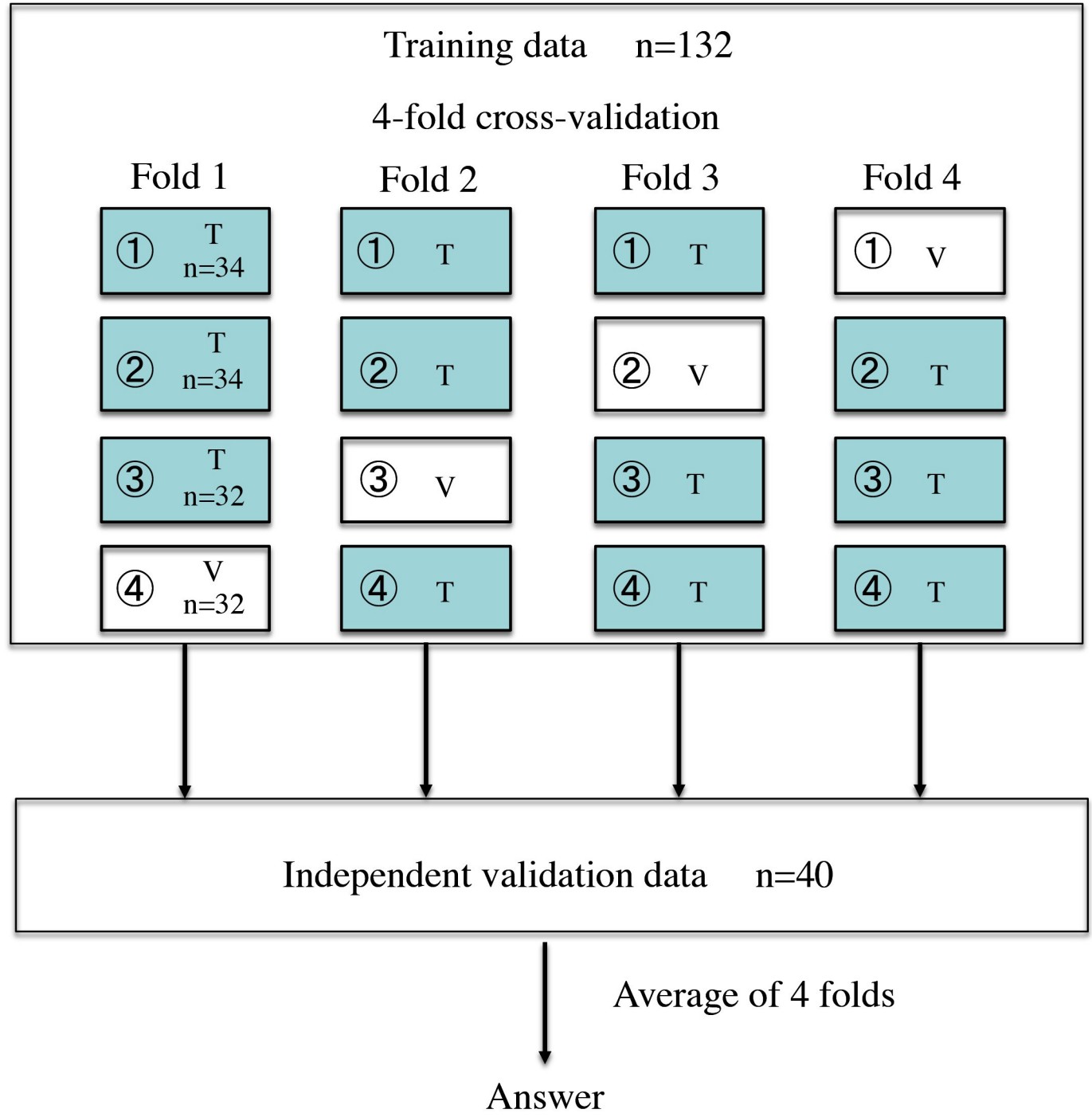

**Fig 3. Training data were divided into four data sets so that the ratio of gastric mucin-positive and gastric mucin-negative lesions did not change, and four-fold cross-validation was performed.** The accuracy of each verification by the four folds was calculated and averaged to the tentative diagnostic accuracy. Using the independent validation data, the diagnostic probability for each case was then calculated in four folds. Finally, the true diagnostic accuracy was obtained by averaging the four diagnostic probabilities. T, Training; V, Validation.

statistically significantly higher than that of all readers because the 95% CI at this time did not overlap with those of any of the readers.

**Table 2. Diagnostic accuracy, area under the curve, and kappa value for each pair of readers.**

| Pairs | Test | Diagnostic accuracy | AUC (95% CI) | ICC (1,1) (95% CI) |
|---|---|---|---|---|
| Pair A (Experienced gynecologic radiologists) | First | 0.650 | 0.720 (0.555–0.885) | 0.666 (0.452–0.808) |
| | Second | 0.725 | 0.814 (0.680–0.949) | |
| Pair B (Young radiologists) | First | 0.775 | 0.777 (0.626–0.927) | 0.750 (0.577–0.859) |
| | Second | 0.700 | 0.765 (0.608–0.923) | |
| Pair C (Gynecologists) | First | 0.625 | 0.812 (0.671–0.953) | 0.680 (0.472–0.816) |
| | Second | 0.700 | 0.759 (0.604–0.914) | |

AUC, area under the curve; CI, confidence interval; ICC, intraclass correlation coefficients

## Discussion

In 2010, Takatsu et al. [9] proposed the 'cosmos pattern' as a characteristic MRI finding of LEGH. In this study, the readers diagnosed LEGH using MR images only, and the highest rate of their positive diagnosis was 77.5%. The precision of the readers was approximately 0.7, indicating that approximately 30% of the GMPLs were judged to be GMNLs. The first reason for judging GMPLs as GMNLs is the presence of GMPLs with atypical imaging findings. In a recent study, the cosmos pattern was found in approximately 60% of GMPLs and was the most frequently observed pattern [10]. In addition, approximately 30% of GMNLs exhibit a cosmos pattern [10]. The second reason is that findings suggestive of LEGH vary from study to study; the definitions of LEGH vary. Takatsu et al. defined the cosmos pattern as 'a pattern of relatively large cysts arranged in the cervical stroma with small cysts or solid components in the centre of lesion' and reported a sensitivity of 87.5% [9]. On the other hand, Ohya et al. defined the cosmos pattern as 'a pattern of small cysts and a solid area in central area with large outer cysts' and reported a higher specificity than sensitivity for GMPLs [10]. Omori et al. also proposed two types of MRI findings of LEGH: flower and raspberry types [16]. To overcome these problems, a common understanding of the definition of LEGH is required.

There are no reports describing differences in diagnostic performance among physicians in LEGH or GMPLs. This study demonstrated that the concordance rate among different physicians' pairs for the diagnosis of GMPLs was not satisfactory (Table 3). Diagnostic concordance was particularly poor between radiologists and gynecologists. The low concordance rate of diagnosis between radiologists and gynecologists suggests that they use different diagnosis methods. Although radiologists are experts in diagnostic imaging, gynecologists make a comprehensive diagnosis based on the clinical symptoms and other clinical information. Since clinical information could not be gathered in this experiment, the accuracy of the gynecologists was considered lower than that of the radiologists. Even among pairs of radiologists, concordance rates are generally not high. We attributed it to two reasons. First, the GMPLs (including LEGH) are difficult to differentiate due to disease rarity and lack of experience. Second, the presence of cases with atypical findings may have been a factor that caused the judgements to vary from day to day, even for the same case. In daily practice, whether or not a patient is

**Table 3. Intraclass correlation coefficient (2, 1) (95% confidence interval) between the physicians' pairs.**

| Pair | Pair B First | Pair B Second | Pair C First | Pair C Second |
|---|---|---|---|---|
| Pair A First | 0.631 (0.401–0.786) | 0.568 (0.321–0.745) | 0.453 (0.083–0.694) | 0.537 (0.066–0.775) |
| Pair A Second | 0.522 (0.255–0.715) | 0.527 (0.267–0.717) | 0.402 (0.068–0.646) | 0.540 (0.082–0.774) |
| Pair B First | | | 0.566 (0.102–0.791) | 0.564 (0.051–0.799) |
| Pair B Second | | | 0.389 (-0.068–0.685) | 0.430 (-0.099–0.745) |

**Table 4. Diagnostic accuracy, area under the curve, and intraclass correlation coefficient (2, 1) value of each convolutional neural network.**

| Test | Tentative diagnostic accuracy | True diagnostic accuracy | AUC (95% CI) | ICC (2,1) value |
|---|---|---|---|---|
| First | 0.588–0.824 | 0.800 | 0.841 (0.711–0.971) | 0.965 |
| Second | 0.677–0.750 | 0.825 | 0.854 (0.731–0.976) | |

AUC, area under the curve; CNN, convolutional neural network; CI, confidence interval; ICC, intraclass correlation coefficient

diagnosed with GMPLs based on MRI findings is highly likely to vary among physicians. This condition poses the risk of being incorrectly diagnosed by MRI diagnosis, resulting in some patients with GMPLs not receiving treatment and others with GMNLs being operated on.

The first possible solution to this problem would be to add diagnosis by cervical mucus, but this method is not covered by insurance in Japan and, therefore, cannot be performed at many facilities. Hence, MRI diagnosis is highly important, but patients in all locations need to have

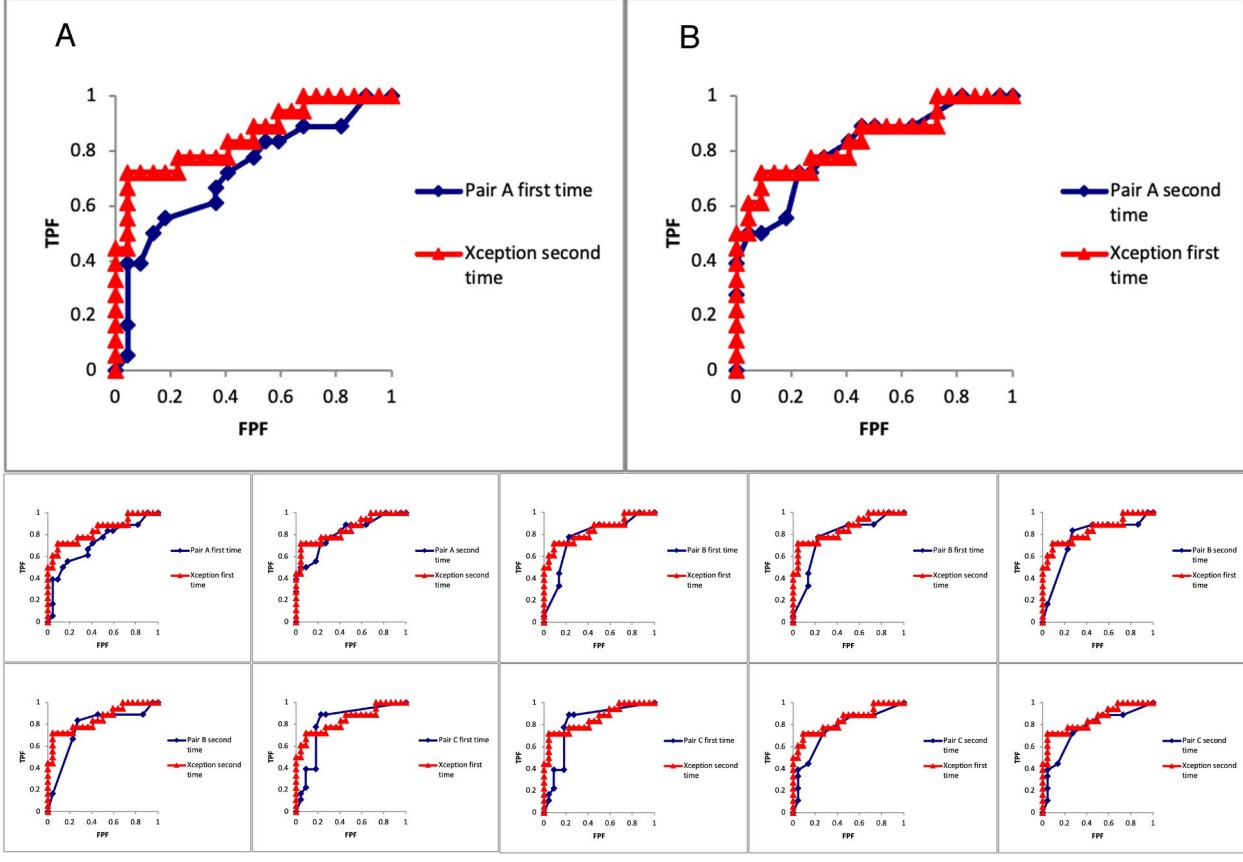

**Fig 4.** (A) Comparison of receiver operating characteristic (ROC) curves at the time of the second independent procedure was entered in Xception and at the time of the first interpretation experiment of pair A with the lowest area under the curve (AUC) among the readers. The AUC at the time of the second independent procedure of Xception was 0.854, which was higher than the AUC at the time of the first interpretation experiment of pair A (0.720). However, there was no statistically significant difference between the two ($p$ = 0.094). (B) Comparison of ROC curves at the time of the first independent procedure of Xception and at the time of the second interpretation experiment of pair A with the highest AUC value. The AUC at the time of the first independent procedure of Xception was 0.841, which was lower than the AUC at the time of the second nterpretation experiment of pair A (0.814). However, there was no statistically significant difference between the two ($p$ = 0.681). Both the first- and second-time AUCs of Xception exceeded those of all readers, but no statistically significant difference in the comparison of ROC curves of all combinations is observed.

equal access to the diagnosis. A possible way to achieve this is through the use of artificial intelligence.

Our results showed that the accuracy rate of Xception was slightly higher than that of the readers. Additionally, the precision, specificity, and F-measure were all higher. However, there is no statistically significant difference in AUC between Xception and readers. These results indicate that Xception and readers have comparable diagnostic performance. The first reason for these results is that the readers did not study the cases in advance and made judgements based on their little clinical experience, whereas the CNN trained 132 cases in advance. Second, the CNN's diagnostics are based on an average of four diagnostic probabilities, which may be responsible for its high diagnostic performance. In addition, Xception showed an extremely high ICC (2,1) value from two independent validations and led to almost identical results. These results indicated that GMPLs diagnosis by the pCNN is as accurate as that by physicians, but with higher reproducibility than that by physicians. This could be one way to solve the previous problems, as diagnosing with Xception could potentially provide the same diagnosis anywhere at the same level as a physician, as long as the same training data are used.

The first limitation of this study is that we included non-surgical and surgical cases. The HIK1083-latex agglutination assay (HIK test), used as the final result, has a very high sensitivity and specificity for detecting gastric-type mucin [13]. However, no test is without false positives and negatives. The result of this study may be influenced by false positives and negatives of HIK test. We could not avoid this problem due to the following reasons. First, our study included cases of GMNLs. Since our institution uses the HIK test, GMNLs are essentially never operated on. If a patient with suspected LEGH is negative for the HIK test, the patient is basically followed up. Second, at least more than 100 cases are necessary to analyze data using the pCNN [14]. It was not possible to examine only surgical cases because LEGH is a rare disease. The second limitation is that it was not possible to analyze what MRI features Xception recognised and diagnosed. Physicians are likely to base their decisions on whether the lesions are in the cosmos pattern or not. However, we averaged the diagnostic probabilities of four-fold cross-validation to avoid overfitting, preventing us from analyzing where Xception focused on the MR image. Finally, the readers conducted image interpretation using only three images for each case. In clinical practice, it is not possible to establish a diagnosis using only three images. Therefore, the diagnostic abilities of the readers might have been underestimated in this study.

In conclusion, our study revealed MRI diagnostic heterogeneity among the readers in distinguishing between GMPLs and GMNLs. Homogeneity in GMPLs and GMNLs diagnosis is important for patients. Xception for learning and diagnosis is as powerful as a physician for distinguishing between GMPLs and GMNLs. If the same training data are used, a highly reproducible diagnosis can be established regardless of facilities. Current problems in the MRI diagnosis of GMPLs and GMNLs may be solved by pCNNs.

## Acknowledgments

We thank Mai Komatsu, Takanori Aonuma, Keisuke Todoroki, Manaka Shinagawa, and Takeuchi Hodaka for conducting image interpretation experiments. We would like to thank Editage (www.editage.com) for English language editing.

## Author Contributions

**Conceptualization:** Ayumi Ohya, Tsutomu Miyamoto.

**Data curation:** Fumihito Ichinohe, Hisanori Kobara.

**Funding acquisition:** Ayumi Ohya, Tsutomu Miyamoto.

**Investigation:** Ayumi Ohya, Fumihito Ichinohe, Hisanori Kobara.

**Methodology:** Ayumi Ohya.

**Project administration:** Tsutomu Miyamoto.

**Resources:** Yasunari Fujinaga.

**Supervision:** Yasunari Fujinaga, Tanri Shiozawa.

**Validation:** Tsutomu Miyamoto, Hisanori Kobara, Tanri Shiozawa.

**Writing – original draft:** Ayumi Ohya.

**Writing – review & editing:** Tsutomu Miyamoto, Yasunari Fujinaga, Tanri Shiozawa.

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
