## [Decision Letter · Decision Letter 0]

4 Jul 2024

PONE-D-24-02686Problems of magnetic resonance diagnosis for gastric-type mucin-positive cervical lesions of the uterus and its solutions using artificial intelligencePLOS ONE

Dear Dr. Miyamoto,

Thank you for submitting your manuscript to PLOS ONE. After careful consideration, we feel that it has merit but does not fully meet PLOS ONE’s publication criteria as it currently stands. Therefore, we invite you to submit a revised version of the manuscript that addresses the points raised during the review process.

We look forward to receiving your revised manuscript.

Kind regards,

Kazunori Nagasaka

Academic Editor

PLOS ONE

Additional Editor Comments:

Dear Authors,

Thank you so much for submitting your manuscript to Plos One.

It is very intriguing study.

Please consider the reviewer's comments and revise the manuscript accordingly.

If you have any inquires, please do not hesitate to contact us.

We look forward to receiving your revised manuscript.

Sincerely,

Plos One

Kazunori Nagasaka

Reviewers' comments:

Reviewer's Responses to Questions

**Comments to the Author**

1. Is the manuscript technically sound, and do the data support the conclusions?

Reviewer #1: Yes

Reviewer #2: Yes

2. Has the statistical analysis been performed appropriately and rigorously? 

Reviewer #1: Yes

Reviewer #2: Yes

3. Have the authors made all data underlying the findings in their manuscript fully available?

Reviewer #1: No

Reviewer #2: Yes

4. Is the manuscript presented in an intelligible fashion and written in standard English?

Reviewer #1: Yes

Reviewer #2: Yes

5. Review Comments to the Author

Reviewer #1: Comments to the Authors

This study used MRI images to differentiate gastric mucin-positive (GMPL) from gastric mucin-negative (GMNL) to validate diagnostic performance and concordance rates by three sets of physicians and a CNN.

The methodology is valid, but there are significant parts that would benefit from substantial revision.

Overall

・In academic or formal writing, especially in scientific papers, "physician" is often preferred over "doctor" when referring to medical doctors.

・The values for metrics such as AUC and ICC are inconsistent; it would be better to standardize them to three decimal places.

・It is confusing because 'validation' and 'test' are used interchangeably; it would be better to use 'test' consistently.

・Independent evaluation of six readers and additional external testing is recommended.

Abstract

・The purpose and conclusion are not aligned. The solution is not clear.

・(axial T2-weighted image [T2WI], axial T1-weighted image [T1WI], and sagittal T2WI)→Please use the same order of notation as in the text.

・I think it is common to express diagnostic ability using AUC.

・Since the same model and the same test data are used, it is natural that the CNN has high reproducibility.

Introduction

・P.5 Line 56, Some reports have described adenocarcinomas in association with LEGH [5–8], does it mean adenocarcinoma other than GAS?

・P.6 Line 1-71, It is redundant and should be concise.

・P.7 Line 87-90, In addition, since artificial intelligence always makes the same diagnosis once it learns, it can be a promising solution when the reproducibility of doctors’ diagnoses is low.→As for CNN, doesn't it make sense to evaluate reproducibility on different test sets?

M&M

MR images

・As Figure 1, patient selection, diagnostic rationale, and assignment to training and testing should be presented in a clear manner.

・P.8 Line 104, ...other than LEGH, LEGH, LEGH with→Isn't there a need for a LEGH in the middle?

・P.8 Line 110, First appearance of BCL is to be spelled out.

・Lesion’s classification should also be tabulated.

・P.10 Line 131, minimum imaging conditions should be described, such as slice thickness, whether 2D or 3D, and the cross-section of imaging (e.g., orthogonal to the cervix).

Diagnostic accuracy and reproducibility by the readers

・Why did you choose to use consensus by pairs? It would be more objective to do each independently and give data for the six people separately.

・What is the definition of diagnostic confidence?

・Is it correct in understanding that they evaluated the same test data with trimming that you put into the CNN?

・Did they evaluate the images without checking for image features that distinguish GMPL from GMNL?

Diagnostic accuracy evaluation by pCNNs

・P.11 Line 139, please list the years of experience of the readers.

・P.11 Line 155, Xception requires a citation

・P.12, Line167, if you are using 3 combined images, I don't think rotation makes sense

Fig1.

Figure 1 is not well explained, but did you use the 3 images as a combined image and each sequence was always trained and tested in the same position and in different colors?

Fig 2.

①②③④ means layers?

I'm not sure, so why don't you change it so we can see which is the same training set, e.g. T①, and leave the current ① out?

Statistical analysis

・P.14 Line 202, Since kappa value comes out of nowhere, it should be explained in the statistics section if you want to bring it out.

Results

・Please describe the characteristics of the patient group (age, etc.) in the first section.

・First, second, ... in Tables 1, 2 and 3 should be unified.

Fig 3.

(A) Comparison of receiver operating characteristic (ROC) curves at the time of the second independent procedure was entered in Xception and at the time of the first interpretation experiment of pair A with the lowest area under the curve (AUC) among the readers.→What made this one so big out of so many?

The former AUC was 0.8535, which was higher than the latter AUC (0.7197).→Please specify what former and latter refer to.

Comparison of ROC curves at the time of the first independent procedure of Xception and　at the time of the second interpretation experiment of pair A with the highest AUC value. →What made this one so big out of so many?

The former AUC was 0.8409, which was lower than the latter AUC (0.8144). →Please specify what former and latter refer to.

Discussion

・P.22, Line 8, How are the diagnostic methods different?

・Please explain why one AUC for an experienced radiologist is lower than for a non-experienced radiologist.

・The value of ICC among experienced radiologists is of interest and should have been evaluated independently for each of the six readers.

P.25, Line 7-8. If the same training data are used, a highly reproducible diagnosis can be established regardless of facilities. →Although a variety of models are included in the training, reproducibility with models not included in the training has not been tested, so this mention cannot be made.

P.25, Line 8-9. Current problems in the MR diagnosis of GMPLs and GMNLs may be solved by pCNNs.→Since no significant difference was found, this reference is also questionable.

Reviewer #2: The focus of your research is excellent. As noted, LEGH and other benign cystic lesions must be distinguished.

Since this is 20 years of data, there are considerable differences in reading due to the imaging accuracy of MRI equipment and other factors.

In addition, it would be helpful to consider at what point the reading differed for cases in which there was a discrepancy in diagnosis in both the Human and CNN groups, to improve the agreement rate.

Although it is unclear how widespread the use of convolutional neural networks will become in the future, the significant difference in the positive diagnosis rate compared to human reading showed that it is a useful tool.

CNN is still not very popular in Japan, but if it becomes popular, it will be applied to preoperative diagnosis of gynecological tumors as well as LEGH, and this is a very interesting paper.

Materials and Methods:

{clinical suspicion of LEGH or GAS (based on ultrasonographic findings and symptoms) }

Please describe in detail what the ultrasound findings and symptoms are specifically.

MR images:

{The types and magnetic field strength of the MR unit used for imaging and the parameters of each sequence varied because the study was retrospective and MR images were acquired over a long period. }

Since the observation period is as long as 20 years and the MRI equipment must have changed many times during that time, I think that the bias by equipment is quite large.

At the very least, please consider how many times the MRI equipment has changed, as well as the details of that equipment and the statistics by equipment.

Diagnostic accuracy and reproducibility by the readers：

{divided into the training (132 patients) and validation (40 patients) test groups}

Please describe the reasons for any differences in the number of patients between groups.

Discussion:

{The second limitation is that it was not possible to analyse what MR features Xception recognised and diagnosed.}

In both Human and CNN groups, please describe the basis for reading LEGH from the MRI images.

Conclusion:

{In conclusion, our study revealed MR diagnostic heterogeneity among the readers in distinguishing between GMPLs and GMNLs.}

I think it is impossible to discuss the concordance rate without considering at what point the differences in reading were observed in the cases with discrepancies in diagnosis in both the Human and CNN groups.

6. PLOS authors have the option to publish the peer review history of their article (what does this mean?). If published, this will include your full peer review and any attached files.

Reviewer #1: No

Reviewer #2: No

---

## [Author Response · Author response to Decision Letter 0]

22 Sep 2024

Response to the reviewers’ comments

We deeply appreciate the reviewers’ educational suggestions. According to the suggestions, we have revised our manuscript as follows. The revised sentences are displayed as red letters in the revised manuscript.

Reviewer #1: Comments to the Authors

This study used MRI images to differentiate gastric mucin-positive (GMPL) from gastric mucin-negative (GMNL) to validate diagnostic performance and concordance rates by three sets of physicians and a CNN.

The methodology is valid, but there are significant parts that would benefit from substantial revision.

Overall

・In academic or formal writing, especially in scientific papers, "physician" is often preferred over "doctor" when referring to medical doctors.

Response: Thank you for pointing this out. We have revised the term “doctor(s)” to “physician(s)” as per your suggestion. (P6L73, P7L83, L86, Table 3 L250, P24L306, L307, P26L337, L338, L340, P27L362 in the revised manuscript)

“vary from doctor to doctor” was revised to “vary among physicians” (P25L319 in the revised manuscript)

・The values for metrics such as AUC and ICC are inconsistent; it would be better to standardize them to three decimal places.

Response: We completely agree with your comment. We have unified the AUC and ICC values to three decimal places. (P4L42, P17L239, L240, L242, Table 2, Table 3, P20L259, L264, Table 4, P21L277, P22L279, L282, L284 in the revised manuscript)

・It is confusing because 'validation' and 'test' are used interchangeably; it would be better to use 'test' consistently.

Response: To avoid confusion among readers, the term “validation” is used in both the text and figure legend.

“validation test” to “validation” (P12L166, P13L171 in the revised manuscript)

“test” to “validation” (P15L203, L205, P16L218 in the revised manuscript)

・Independent evaluation of six readers and additional external testing is recommended.

Response: As per your suggestion, we conducted additional experiments. The results demonstrated a diagnostic accuracy of up to 80% and an AUC of up to 0.859. Diagnostic accuracy of the readers was slightly lower than that of CNN, and AUC of the readers was slightly higher than that of CNN; however, the 95% confidence intervals overlapped and were not significantly different. The following are the results of additional experiments.

Reader Diagnostic accuracy AUC (95% CI)

A 0.75 0.859 (0.746–0.972)

B 0.7 0.797 (0.658–0.936)

C 0.675 0.705 (0.540–0.869)

D 0.8 0.816 (0.678–0.953)

E 0.775 0.793 (0.646–0.940)

F 0.7 0.729 (0.566–0.891)

Overall, both the diagnostic accuracy and AUC slightly improved when the reading experiment was conducted individually compared to when it was performed in pairs. However, the diagnostic accuracy of the individual is slightly lower than that of the CNN, and the individual AUC is not statistically significantly different from the CNN AUC. The same is the case for the pair.

Abstract

・The purpose and conclusion are not aligned. The solution is not clear.

Response: Based on your valuable comment, we have aligned our objectives and conclusions. The first sentence of the conclusion has been changed as follows: Variation in the inter-reader or intra-reader accuracy in MRI diagnosis limits differentiation between GMPL and GMNL. (P4L43-44 in the revised manuscript)

・(axial T2-weighted image [T2WI], axial T1-weighted image [T1WI], and sagittal T2WI)→Please use the same order of notation as in the text.

Response: As per your comment, we have ensured the dame order of notation as in the text. (P3L31-32 in the revised manuscript)

・I think it is common to express diagnostic ability using AUC.

Response: We completely agree with your comment. Therefore, the first sentence of the conclusion has been changed as follows: Variation in the inter-reader or intra-reader accuracy in MRI diagnosis limits differentiation between GMPL and GMNL. (P4L43-44 in the revised manuscript)

・Since the same model and the same test data are used, it is natural that the CNN has high reproducibility.

Response: In CNN, the same model and the same test data are not necessarily reproducible. This is because even if the CNN uses the same model, different training methods may lead to different test results. Many papers do not confirm the reproducibility of the results by CNN. Thus, we have demonstrated that CNN is highly reproducible, even if it trains on the same data in different ways.

Introduction

・P.5 Line 56, Some reports have described adenocarcinomas in association with LEGH [5–8], does it mean adenocarcinoma other than GAS?

Response: We greatly appreciate your point of view. We have deleted that sentence. (P5L53 in the revised manuscript)

・P.6 Line 1-71, It is redundant and should be concise.

Response: Thank you for your suggestion. 

“Various benign lesions with cystic components occur in the uterine cervix. Common benign lesions include Nabothian cysts, tunnel clusters, lobular endocervical glandular hyperplasia (LEGH), endometriosis, and cervical polyps [1]. LEGH is a hyperplastic benign lesion first proposed by Nucci et al. [2], but it may be a precursor lesion for gastric-type mucinous carcinomas (GAS) with a very poor prognosis [3,4]. Some reports have described adenocarcinomas in association with LEGH [5–8]. The most recent study demonstrated that malignant change of LEGH occurs at a frequency of 1.4% [9], but frequent and long-term follow-up or surgical treatment is selected once LEGH is diagnosed. In contrast, other benign cystic lesions of the uterine cervix do not require follow-up as frequently as LEGH. Therefore, clinically, LEGH and other benign cystic lesions must be distinguished.

LEGH has been reported to exhibit magnetic resonance imaging (MRI) findings called ‘cosmos pattern’, while Nabothian cysts show coarse cysts pattern [10]. However, some Nabothian cysts exhibit MR findings similar to those of LEGH [11]. The decisive difference from other benign lesions is that LEGH and GAS exhibit a distinctive pyloric gland metaplasia and secrete gastric-type mucin, which has O-linked oligosaccharides with a terminal α1,4-linked N-acetylglucosamine (αGlcNAc) residue [12]. Because gastric-type mucin is a neutral mucin, LEGH and GAS exhibit a ‘two-colour pattern’ on Pap smears [13]. Further, αGlcNAc expression exhibits positive immunohistochemical findings for monoclonal antibody HIK1083. αGlcNAc has been detected in cervical mucus by latex agglutination assay using monoclonal antibody HIK1083 [14].” was revised to 

“Common benign lesions in the uterine cervix include Nabothian cysts, tunnel clusters, lobular endocervical glandular hyperplasia (LEGH), endometriosis, and cervical polyps [1]. LEGH is a benign lesion first proposed by Nucci et al. [2], but it may be a precursor lesion for gastric-type mucinous carcinomas (GAS) [3–8]. Therefore, frequent and long-term follow-up or surgical treatment is selected once LEGH is diagnosed. In contrast, other benign cystic lesions of the uterine cervix do not require follow-up as frequently as LEGH. Therefore, clinically, LEGH and other benign cystic lesions must be distinguished.

LEGH shows magnetic resonance imaging (MRI) findings called ‘cosmos pattern’, while Nabothian cysts show coarse cysts pattern [9]. However, some Nabothian cysts exhibit MR findings similar to those of LEGH [10]. The decisive difference from other benign lesions is that LEGH and GAS secrete gastric-type mucin, which has O-linked oligosaccharides with a terminal α1,4-linked N-acetylglucosamine (αGlcNAc) residue [11]. Because gastric-type mucin is a neutral mucin, LEGH and GAS exhibit a ‘two-color pattern’ on Pap smears [12]. Further, αGlcNAc has been detected in cervical mucus by latex agglutination assay using monoclonal antibody HIK1083 [13].” (P5L50-P6L65 in the revised manuscript)

・P.7 Line 87-90, In addition, since artificial intelligence always makes the same diagnosis once it learns, it can be a promising solution when the reproducibility of doctors’ diagnoses is low.→As for CNN, doesn't it make sense to evaluate reproducibility on different test sets?

Response: Thank you for your comments. Physicians used the same validation data to determine reproducibility. The CNN would have to be evaluated under the same conditions to be comparable. In addition, because GMPLs are relatively rare diseases, it has been impossible to provide separate validation datasets.

M&M

MR images

・As Figure 1, patient selection, diagnostic rationale, and assignment to training and testing should be presented in a clear manner.

Response: We completely agree with your comment. We have added a new Figure 1 and its caption to make patient selection, diagnostic rationale, and assignment to training and testing easier to understand. Therefore, Figure numbers were rearranged. “The process of patient selection, and grouping is summarized in Fig 1.” was added. (P7L95-96 in the revised manuscript)

・P.8 Line 104, ...other than LEGH, LEGH, LEGH with→Isn't there a need for a LEGH in the middle?

Response: Yes, there is. For clarity, “(BCL)” was added after “other than LEGH”. (P8L101 in the revised manuscript)

・P.8 Line 110, First appearance of BCL is to be spelled out.

Response: As noted above, “BCL” is already exists at P8L101.

・Lesion’s classification should also be tabulated.

Response: As per your suggestion, we have summarized the pathological features of the lesion in new Table 1. Therefore, we rearranged Table numbers.

“The classification of histopathologic lesions is summarized in Table 1.” was added at P9L113-114 in the revised manuscript.

・P.10 Line 131, minimum imaging conditions should be described, such as slice thickness, whether 2D or 3D, and the cross-section of imaging (e.g., orthogonal to the cervix).

Response: As you have accurately pointed out, we have added basic information about the image as follows: All images have a slice thickness of 2 to 7.5 mm and are captured as two-dimensional images. (P12L161-162 in the revised manuscript)

Diagnostic accuracy and reproducibility by the readers

・Why did you choose to use consensus by pairs? It would be more objective to do each independently and give data for the six people separately.

Response: The CNN performs validation using four differently trained folds. Therefore, we considered that humans should also be examined by more than one group to be considered as equals, so we examined them in pairs. Additional independent studies were conducted for each of the six individuals.

・What is the definition of diagnostic confidence?

Response: The confidence level of the diagnosis was stated as a percentage based on each physician's experience. We have appended this statement as follows: The confidence level of the diagnosis was stated as a percentage based on each physician’s experience. (P13L173-174 in the revised manuscript)

・Is it correct in understanding that they evaluated the same test data with trimming that you put into the CNN?

Response: Yes, it is. 

・Did they evaluate the images without checking for image features that distinguish GMPL from GMNL?

Response: Yes, they did. They did not have information of useful image features that distinguished GMPL from GMLN when they evaluated the images.

Diagnostic accuracy evaluation by pCNNs

・P.11 Line 139, please list the years of experience of the readers.

Response: As per your comment, we have added the years of experience of each reader. (P12L169-P13L170 in the revised manuscript)

・P.11 Line 155, Xception requires a citation

Response: As per your comment, we have added the paper of Xception to the references (ref# 15). 

・P.12, Line167, if you are using 3 combined images, I don't think rotation makes sense

Response: When the three images are superimposed, the angle of rotation is limited to 10° at random. Rotation is not unnatural because the cervix is tilted differently in each person. This method is common and used previous reports (La Grace Saint-Esteven A, et al. Comput Biol Med. 2022 142: 105215).

Fig1.

Figure 1 is not well explained, but did you use the 3 images as a combined image and each sequence was always trained and tested in the same position and in different colors?

Response: CNN recognizes three images by inputting them into three color channels. We input a T2-weighted sagittal section in the red channel, a T2-weighted transverse section in the green channel, and a T1-weighted transverse section in the blue channel. We ensured that the images were always entered in the same way during training and during validation.

Fig 2.

①②③④ means layers?

I'm not sure, so why don't you change it so we can see which is the same training set, e.g. T①, and leave the current ① out?

Response: ①②③④　are the training data randomly divided into four parts; of the four parts, three colored data are used for training and the uncolored data are used for provisional validation. CNN training uses three of these four divisions of data to train four different types of training.

Statistical analysis

・P.14 Line 202, Since kappa value comes out of nowhere, it should be explained in the statistics section if you want to bring it out.

Response: Thank you for pointing this out. This is an error. We have corrected “kappa value” to “ICC (1) or (2)”. (P16L225 in the revised manuscript)

Results

・Please describe the characteristics of the patient group (age, etc.) in the first section.

Response: We are unable to describe the characteristics of the patients in the Results section because they are not part of the survey. The age of the patients is described in the “Patient population” section as follows: Patients ranged in age from 26 to 82 years, with an average age of 48.7 years. (P7L94-95 in the revised manuscript)

・First, second, ... in Tables 1, 2 and 3 should be unified.

Response: As per your comments, we have changed Table 3 to unify them.

Fig 3.

Comparison of receiver operating characteristic (ROC) curves at the time of the second independent procedure was entered in Xception and at the time of the first interpretation experiment of pair A with the lowest area under the curve (AUC) among the readers.→What made this one so big out of so many?

Response: We selected and enlarged this graph because we compared the highest AUC in Xception to the lowest AUC for the physician pair.

The former AUC was 0.8535, which was higher than the latter AUC (0.7197).→Please specify what former and latter refer to.

Response: Thank you for pointing this out. It was difficult to understand, so we have corrected the relevant part as follows: The AUC at the time of second independent procedure of Xception was 0.854, which was higher than the AUC at the time of the first interpretation experiment of pair A (0.720) (P21L277-P22L279 in the revised manuscript)

Comparison of ROC curves at the time of the first independent procedure of Xception and at the time of the second interpretation experiment of pair A with the highest AUC value. →What made this one so big out of so many?

Response: We selected and enlarged this graph because we compared the lowest AUC in Xception to the highest AUC for the physician pair.

The former AUC was 0.8409, which was lower than the latter AUC (0.8144). →Please specify what former and latter refer to.

Response: Thank you for pointing this out. To make the sentence easier to understand, we have corrected the relevant part as follows: The AUC at the time of the first independent procedure of Xception was 0.841, which was lower than the AUC at the time of the second interpretation experiment of pair A (0.814). (P22L282-284 in the revised manuscript)

Discussion

・P.22, Line 8, How are the diagnostic methods different?

Response: We have added the following sentences. “Although radiologists are experts in diagnostic imaging, gynecologists make a comprehensive diagnosis based on the clinical symptoms and other clinical information. Since clinical information could not be gathered in this experiment, the accuracy of the gynecologists was considered lower than that of the radiologists.” (P24L310-314 in the revised manuscript)

・Please explain why one AUC for an experienced radiologist is l

---

## [Decision Letter · Decision Letter 1]

4 Oct 2024

PONE-D-24-02686R1Problems of magnetic resonance diagnosis for gastric-type mucin-positive cervical lesions of the uterus and its solutions using artificial intelligencePLOS ONE

Dear Dr. Miyamoto,

Thank you for submitting your manuscript to PLOS ONE. After careful consideration, we feel that it has merit but does not fully meet PLOS ONE’s publication criteria as it currently stands. Therefore, we invite you to submit a revised version of the manuscript that addresses the points raised during the review process.

We look forward to receiving your revised manuscript.

Kind regards,

Kazunori Nagasaka

Academic Editor

PLOS ONE

Journal Requirements:

Additional Editor Comments:

Dear Authors,

Thank you so much for your submission to Plos One.

As the reviewers pointed out in their comments, please revise the manuscript accordingly.

Their primary focus would be to increase your impact on your claim in the manuscript.

We look forward to your revised manuscript soon.

Sincerly,

Kazunori Nagasaka

Reviewers' comments:

Reviewer's Responses to Questions

**Comments to the Author**

1. If the authors have adequately addressed your comments raised in a previous round of review and you feel that this manuscript is now acceptable for publication, you may indicate that here to bypass the “Comments to the Author” section, enter your conflict of interest statement in the “Confidential to Editor” section, and submit your "Accept" recommendation.

Reviewer #1: All comments have been addressed

Reviewer #2: All comments have been addressed

2. Is the manuscript technically sound, and do the data support the conclusions?

Reviewer #1: Yes

Reviewer #2: Yes

3. Has the statistical analysis been performed appropriately and rigorously? 

Reviewer #1: Yes

Reviewer #2: Yes

4. Have the authors made all data underlying the findings in their manuscript fully available?

Reviewer #1: Yes

Reviewer #2: Yes

5. Is the manuscript presented in an intelligible fashion and written in standard English?

Reviewer #1: Yes

Reviewer #2: Yes

6. Review Comments to the Author

Reviewer #1: Thank you for the revision according to the comments. I understand that there is no significant difference in the results whether in pairs or individually.

Please consider the following minor revisions.

Abstract

P.3, line 36, (sagittal T2WI and axial T2WI/T1WI) may be better.

Overall: Since "magnetic resonance imaging (MRI) findings" is mentioned in Introduction p.5, line 58, I think it is better to use "MRI findings", "MRI diagnosis", and "MR unit" instead of "MR findings", "MR diagnosis", and "MRI unit" thereafter. However, "MR images" can remain as is.

Table 1 should include other BCLs with histologically unconfirmed diagnosis, as well as information on the number of patients and their ages. The title should be changed accordingly.

MR images

P.12, L.168, For the CNN, it is assumed that one image per patient was used, so it would be better to add "per patient" as in: "One image showing the maximum cross-section of the lesion was selected from each of these three sequences per patient, and the three images were used for analysis."

Diagnostic accuracy and reproducibility by the readers

P.12, L168, GMNL. (Fig 1)→Move the period to the end.

Diagnostic accuracy evaluation by pCNNs

P.14, L189, the validation group. (Fig 1)→Move the period to the end.

P.14, L200, I think "(Fig 2)" can be omitted from "Using the training data, all three images were input into Xception."

Fig2. "Input" is used for converting Fig into color, and it may cause confusion with "Input" for CNN in the main text. Therefore, I think it would be better to change the text in the figure to something like "Conversion."

Fig3. The "test" in the figure and "the validation data" in the figure description may cause confusion with cross-validation, so I think it would be better to unify them as "independent validation".

The first letters in the abbreviation explanations of Fig 3 should be consistently either capitalized or in lowercase.

Diagnostic accuracy evaluation by pCNNs

P.15, L203-205

“Finally, images of the validation data were input into the model trained in each fold, and the true diagnostic accuracy and AUC for the validation data were determined by averaging the diagnostic probabilities by the four models (Fig 3).”

I think using "independent validation" would make the above sentences clearer.

Table 3. The notation in the "Second" column of Pair B is incorrect.

In the Figure legends of Fig 4, please standardize the decimal places for the p-values to three digits, same as the others. “However, there was no statistically significant difference between the two (p =

0.0939).” “However, there was no statistically significant difference between the two (p =

0.6813).”

Table 4. It would be better to include the meaning of "Tentative diagnostic accuracy" in the main text.

Please change the lower limit value of the Tentative diagnostic accuracy for "second" to three decimal places.

Discussion

P.25. L.327-331. Since "readers" appears consecutively, I think the second "readers" can be omitted.

Reviewer #2: Since CNN is almost as accurate as the reader, it has not shown usefulness in reading at this time. It has little IMPACT as a new FINDINGS. In addition, CNN may have the potential to improve diagnostic reproducibility, but in actual clinical practice, when a treatment plan is decided based on MRI readings, “diagnostic reproducibility” is not as useful as “diagnostic accuracy. I don't feel it is useful.

What would you estimate the number of LEGH clinical experiences of the readers to be for a CNN that has trained 132 LEGH cases? For example, I think it is much more experienced than the group of young radiologists. What do you consider the difference?

As Reviewer 1 also pointed out, I don't understand the significance of examining the pairs. I think a comparison between individuals would be better; you have also disclosed the results between individuals in Response, and this should be noted and discussed.

The group of readers did not study the LEGH cases in advance and made decisions based on their limited clinical experience, which may be a disadvantage compared to the CNN, which is supposed to be studied in advance.

Isn't it a leap to conclude that a CNN's finding of a significant difference in diagnostic reproducibility can provide the same level of diagnosis everywhere as a reading physician, when no significant difference in diagnostic accuracy has been found?

Without analysis of what MRI features CNNs recognized and diagnosed, I don't think this will lead to the widespread use and improvement of CNNs in the future.

7. PLOS authors have the option to publish the peer review history of their article (what does this mean?). If published, this will include your full peer review and any attached files.

Reviewer #1: No

Reviewer #2: No

---

## [Author Response · Author response to Decision Letter 1]

18 Nov 2024

Response to the reviewers’ comments

We deeply appreciate the reviewers’ educational suggestions. According to the suggestions, we have revised our manuscript as follows. The revised sentences are displayed as red letters in the revised manuscript.

Reviewer #1: 

Thank you for the revision according to the comments. I understand that there is no significant difference in the results whether in pairs or individually.

Please consider the following minor revisions.

Abstract

P.3, line 36, (sagittal T2WI and axial T2WI/T1WI) may be better.

Response: Thank you for pointing this out. We have corrected it, as you suggested.

Overall: Since "magnetic resonance imaging (MRI) findings" is mentioned in Introduction p.5, line 58, I think it is better to use "MRI findings", "MRI diagnosis", and "MR unit" instead of "MR findings", "MR diagnosis", and "MRI unit" thereafter. However, "MR images" can remain as is.

Response: Thank you for pointing this out. We have corrected it, as you suggested.

Table 1 should include other BCLs with histologically unconfirmed diagnosis, as well as information on the number of patients and their ages. The title should be changed accordingly.

Response: Thank you for your feedback. We have revised Table 1 and its title, as you suggested.

MR images

P.12, L.168, For the CNN, it is assumed that one image per patient was used, so it would be better to add "per patient" as in: "One image showing the maximum cross-section of the lesion was selected from each of these three sequences per patient, and the three images were used for analysis."

Response: Thank you for pointing this out. We have added “per patient”, as you suggested.

Diagnostic accuracy and reproducibility by the readers

P.12, L168, GMNL. (Fig 1)→Move the period to the end.

Response: We apologize for our careless mistake. We have corrected it, as you suggested.

Diagnostic accuracy evaluation by pCNNs

P.14, L189, the validation group. (Fig 1)→Move the period to the end.

Response: We apologize for our careless mistake. We have corrected it, as you suggested.

P.14, L200, I think "(Fig 2)" can be omitted from "Using the training data, all three images were input into Xception."

Response: Thank you for pointing this out. We have corrected it, as you suggested.

Fig2. "Input" is used for converting Fig into color, and it may cause confusion with "Input" for CNN in the main text. Therefore, I think it would be better to change the text in the figure to something like "Conversion."

Response: Thank you for your suggestion. As you said, Fig 2 shows that a black and white images are inputted and consequently converted into color images. We have revised Fig 2, as you suggested. In addition, we have revised the Figure Legends of Fig 2 as follows.

“A sagittal T2-weighted image was input into the red channel, an axial T2-weighted image was input into the green channel, and a T1-weighted image was input into the blue channel. These converted images were processed to include only the lesion and cervical stroma of the uterus.” (P16 L214-217 in the revised manuscript)

Fig3. The "test" in the figure and "the validation data" in the figure description may cause confusion with cross-validation, so I think it would be better to unify them as "independent validation".

The first letters in the abbreviation explanations of Fig 3 should be consistently either capitalized or in lowercase.

Response: According to your suggestion, we have revised Fig 3, and its Figure legends.

Diagnostic accuracy evaluation by pCNNs

P.15, L203-205

“Finally, images of the validation data were input into the model trained in each fold, and the true diagnostic accuracy and AUC for the validation data were determined by averaging the diagnostic probabilities by the four models (Fig 3).”

I think using "independent validation" would make the above sentences clearer.

Response: Thank you for your feedback. Your point is certainly valid. According to your suggestion, we have added “independent” at P15L207 and L208 in the revised manuscript.

Table 3. The notation in the "Second" column of Pair B is incorrect.

Response: We apologize for our careless mistake. We have corrected it.

In the Figure legends of Fig 4, please standardize the decimal places for the p-values to three digits, same as the others. “However, there was no statistically significant difference between the two (p =0.0939).” “However, there was no statistically significant difference between the two (p =0.6813).”

Response: According to your suggestion, we have revised 0.0939 to 0.094, and 0.6813 to 0.681. (L285 and L290 in the revised manuscript)

Table 4. It would be better to include the meaning of "Tentative diagnostic accuracy" in the main text.

Response: Thanks for your suggestion. We have corrected the sentence on P14L204-206 in the revised manuscript as follows: The accuracy of each verification by the four-fold cross-validation was calculated and averaged to the tentative diagnostic accuracy, which indicates degree of training.

Please change the lower limit value of the Tentative diagnostic accuracy for "second" to three decimal places.

Response: Thanks for your point of view. We have corrected the relevant area.

Discussion

P.25. L.327-331. Since "readers" appears consecutively, I think the second "readers" can be omitted.

Response: According to your suggestion, “Additionally, the precision, specificity, and F-measure were all higher than those of the readers” was revised to “Additionally, the precision, specificity, and F-measure were all higher.” (P26L333 in the revised manuscript)

Reviewer #2: 

Since CNN is almost as accurate as the reader, it has not shown usefulness in reading at this time. It has little IMPACT as a new FINDINGS. In addition, CNN may have the potential to improve diagnostic reproducibility, but in actual clinical practice, when a treatment plan is decided based on MRI readings, “diagnostic reproducibility” is not as useful as “diagnostic accuracy. I don't feel it is useful.

Response: We understand your point of view. However, this study aims to explore the issues in the diagnosis of LEGH, demonstrating that current human MRI diagnoses of LEGH are inconsistent and that the interpretation results can vary significantly depending on the reader. Diagnosis by the CNN was only used as a comparison to verify the variability of human diagnosis, and it is not the purpose of this study to verify the superiority of CNN. We agree that the diagnostic accuracy is important. However, we believe that the reproducibility is equally important, as low reproducibility includes the possibility that accuracy may be reduced if a different doctor diagnoses the patient. 

What would you estimate the number of LEGH clinical experiences of the readers to be for a CNN that has trained 132 LEGH cases? For example, I think it is much more experienced than the group of young radiologists. What do you consider the difference?

Response: There is a misunderstanding in your interpretation. A CNN cannot provide an adequate diagnosis unless it is trained on around 100 cases. On the other hand, the 'young radiologist' in this study, though less experienced with LEGH, is a specialist in diagnostic imaging. The fact that the pairs of young radiologists achieved the highest diagnostic accuracy cannot be ignored. Since there was no significant difference in accuracy between the CNN and the radiologists, it can be concluded that even though the young radiologists did not have prior training, they were not inferior to the CNN.　

As Reviewer 1 also pointed out, I don't understand the significance of examining the pairs. I think a comparison between individuals would be better; you have also disclosed the results between individuals in Response, and this should be noted and discussed.

Response: In response to Reviewer 1’s comments, we disclosed the individual physicians’ reading results, but there was no difference compared to when the readings were done in pairs. Therefore, we did not include it in the manuscript. Additionally, the CNN provides final results by averaging the outcomes of the four folds. Since CNN is using an average of 4 folds to answer the question, we thought the physicians should also be in the group.

The group of readers did not study the LEGH cases in advance and made decisions based on their limited clinical experience, which may be a disadvantage compared to the CNN, which is supposed to be studied in advance.

Response: We agree with your comment. If the readers had been trained beforehand, they might have had a better accuracy. However, as mentioned above, diagnosis by the CNN was only used as a comparison to verify the variability of human diagnosis, and it is not the purpose of this study to verify the superiority of CNN. 

Isn't it a leap to conclude that a CNN's finding of a significant difference in diagnostic reproducibility can provide the same level of diagnosis everywhere as a reading physician, when no significant difference in diagnostic accuracy has been found?

Response: What we would like to say is that when a certain finding was found, humans may vary in their diagnosis, but CNN do not, making them useful as a criterion for diagnosis. It goes without saying that a high accuracy and reproducibility are ideal.

Without analysis of what MRI features CNNs recognized and diagnosed, I don't think this will lead to the widespread use and improvement of CNNs in the future.

Response: Thank you for your important comment. We agree your comment. We also believe it is important to clarify what features of MRI images the CNN captures and understands for its development. However, the specific details of how the CNN analyzes images are not clearly established, so it is even unclear if it operates in the same way as human interpretation. Additionally, in this study, the CNN uses the average results of the four folds as the final outcome, and due to the low agreement rate among physicians across the three groups, the analysis and comparative evaluation are very complex and challenging. We think that exploring how the CNN recognizes MRI images needs to be addressed through a different approach and should be a focus for future studies.

---

## [Decision Letter · Decision Letter 2]

3 Dec 2024

Problems of magnetic resonance diagnosis for gastric-type mucin-positive cervical lesions of the uterus and its solutions using artificial intelligence

PONE-D-24-02686R2

Dear Dr. Miyamoto,

We’re pleased to inform you that your manuscript has been judged scientifically suitable for publication and will be formally accepted for publication once it meets all outstanding technical requirements.

Kind regards,

Kazunori Nagasaka

Academic Editor

PLOS ONE

Additional Editor Comments (optional):

Dear Authors,

Thank you for submitting your manuscript to Plos One.

Our reviewers suggested the manuscript is ready to accept for publication.

Congulatulatin on your works and we look forward to receiving your future studies!!

Sincerely,

Kazunori Nagasaka

Reviewers' comments:

Reviewer's Responses to Questions

**Comments to the Author**

1. If the authors have adequately addressed your comments raised in a previous round of review and you feel that this manuscript is now acceptable for publication, you may indicate that here to bypass the “Comments to the Author” section, enter your conflict of interest statement in the “Confidential to Editor” section, and submit your "Accept" recommendation.

Reviewer #1: (No Response)

Reviewer #2: All comments have been addressed

2. Is the manuscript technically sound, and do the data support the conclusions?

Reviewer #1: (No Response)

Reviewer #2: Yes

3. Has the statistical analysis been performed appropriately and rigorously? 

Reviewer #1: (No Response)

Reviewer #2: Yes

4. Have the authors made all data underlying the findings in their manuscript fully available?

Reviewer #1: (No Response)

Reviewer #2: Yes

5. Is the manuscript presented in an intelligible fashion and written in standard English?

Reviewer #1: (No Response)

Reviewer #2: Yes

6. Review Comments to the Author

Reviewer #1: (No Response)

Reviewer #2: Thank you for the revision according to the comments. I have understood that you used CNN as comparators to validate the variability of human diagnoses, and it is not the purpose of this study to verify the superiority of CNN.

I now understand why you considered pairs instead of individuals.

I have accepted your manuscript for publication.

7. PLOS authors have the option to publish the peer review history of their article (what does this mean?). If published, this will include your full peer review and any attached files.

Reviewer #1: No

Reviewer #2: No

---

## [Editor Report · Acceptance letter]

16 Dec 2024

PONE-D-24-02686R2 

PLOS ONE

Dear Dr. Miyamoto, 

I'm pleased to inform you that your manuscript has been deemed suitable for publication in PLOS ONE. Congratulations! Your manuscript is now being handed over to our production team.

Kind regards, 

on behalf of

Professor Kazunori Nagasaka 

Academic Editor

PLOS ONE